# ADASPLASH: Adaptive Sparse Flash Attention

**Nuno Gonçalves**[1]   **Marcos Treviso**[2]   **André F. T. Martins**[1 2 3]

## Abstract

The computational cost of softmax-based attention in transformers limits their applicability to long-context tasks. Adaptive sparsity, of which $\alpha$-entmax attention is an example, offers a flexible data-dependent alternative, but existing implementations are inefficient and do not leverage the sparsity to obtain runtime and memory gains. In this work, we propose ADASPLASH, which combines the efficiency of GPU-optimized algorithms with the sparsity benefits of $\alpha$-entmax. We first introduce a hybrid Halley-bisection algorithm, resulting in a 7-fold reduction in the number of iterations needed to compute the $\alpha$-entmax transformation. Then, we implement custom Triton kernels to efficiently handle adaptive sparsity. Experiments with RoBERTa and ModernBERT for text classification and single-vector retrieval, along with GPT-2 for language modeling, show that our method achieves substantial improvements in runtime and memory efficiency compared to existing $\alpha$-entmax implementations. It approaches—and in some cases surpasses—the efficiency of highly optimized softmax implementations like FlashAttention-2, enabling long-context training while maintaining strong task performance.[1]

## 1. Introduction

Central to the success of transformers (Vaswani et al., 2017) lies the attention mechanism, where each token in a sequence attends directly to every other token. Attention probabilities are computed through the **softmax** transformation, which always assigns a nonzero probability to every token. However, for long context inputs, the accumulation of small probabilities can lead to dispersion (Veličković et al., 2025).

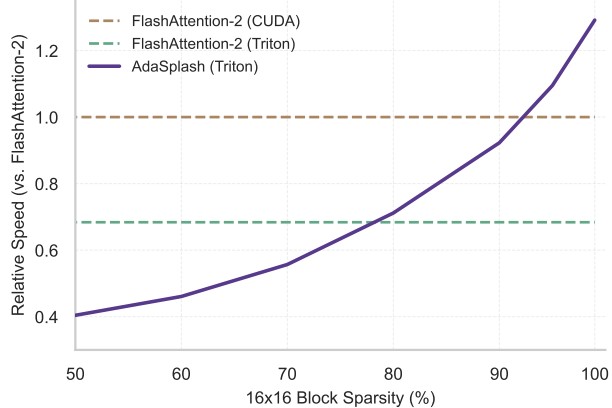

*Figure 1.* Runtime (Fwd+Bwd) as a function of input sparsity for non-causal attention. While the highly-optimized FlashAttention-2 maintains a constant runtime across varying levels of sparsity, ADASPLASH effectively leverages sparsity to obtain speed-ups, eventually outperforming FlashAttention-2 as sparsity grows.

In fact, previous research shows that attention probabilities tend to peak around a small number of tokens (Voita et al., 2019; Treviso et al., 2022), which suggests that model performance and computational efficiency can be increased by leveraging attention sparsity. This has motivated methods that predefine sparse masks (Beltagy et al., 2020; Zaheer et al., 2020b), rely on clustering-based strategies (Kitaev et al., 2020), or low-rank approximate attention (Choromanski et al., 2021; Peng et al., 2021; Xiong et al., 2021; Chen et al., 2021). Some of these techniques show the potential of sparsity to mitigate memory and computation bottlenecks, but they often require architectural modifications or crude approximations, limiting their flexibility and generality.

A related line of research explores adaptive and differentiable sparse activations as surrogates of softmax, such as **sparsemax** (Martins & Astudillo, 2016) and, more broadly, the $\alpha$-**entmax** family (Peters et al., 2019; Correia et al., 2019). By assigning zero probability to irrelevant tokens, these activations eliminate their residual influence, reducing the dilution of attention scores and potentially improving both performance and interpretability. Unfortunately, existing algorithms and implementations for these adaptive sparse activations do not exploit the sparsity, being slower than softmax-based attention and struggling to scale

[1]Instituto Superior Técnico, Universidade de Lisboa, Portugal [2]Instituto de Telecomunicações, Lisbon, Portugal [3]Unbabel, Lisbon, Portugal. Correspondence to: Nuno Gonçalves <nuno.m.goncalves@tecnico.ulisboa.pt>.

*Proceedings of the $42^{nd}$ International Conference on Machine Learning*, Vancouver, Canada. PMLR 267, 2025. Copyright 2025 by the author(s).

[1]Code: https://github.com/deep-spin/adasplash

effectively with context length, primarily due to the lack of hardware-optimized implementations like FlashAttention-2 (Dao, 2024) or support from programming models like FlexAttention (Dong et al., 2024).

This paper addresses this problem by providing new algorithms and implementations to improve the computational efficiency of the family of $\alpha$-entmax activations. Our main contributions include a faster and GPU-friendly algorithm for calculating $\alpha$-entmax, alongside a Triton kernel (Tillet et al., 2019) for computing entmax-based attention, which we call ADASPLASH. In particular, ADASPLASH advances the goal of supporting training of adaptively sparse models with longer context lengths, as shown in Figure 1. We demonstrate the potential and scalability of our approach through experiments with synthetic data and with several natural language processing benchmarks for encoder-only and decoder-only models, achieving substantial improvements over previous $\alpha$-entmax implementations and approaching (sometimes surpassing) the efficiency of softmax-based attention with FlashAttention-2, with strong performance on downstream tasks.

## 2. Background

### 2.1. Hardware Performance

Modern GPUs, such as the Nvidia H100, are designed for efficient parallel computation using a hierarchical memory architecture, with high-bandwidth memory (HBM) providing large capacity but slower access compared to the smaller, faster on-chip SRAM. Efficient use of SRAM is critical to minimize the memory bottlenecks caused by frequent HBM accesses. GPUs execute operations (kernels) via thousands of threads organized into thread blocks, where data is loaded from HBM into SRAM for computation before being written back. Kernel fusion is a key optimization strategy that combines multiple operations into a single kernel, reducing intermediate HBM accesses by directly computing and storing final results. While compilers like `torch.compile` can automate fusion for simple operations (Ansel et al., 2024), complex tasks such as attention mechanisms require custom strategies to reorder operations and optimize memory usage effectively. Our method leverages this GPU memory organization by implementing block-wise computations, recomputation strategies, and kernel fusion specifically tailored for sparse attention, as detailed in §3.2.1 and §3.2.2.

### 2.2. Standard Attention

Given a set of matrices $\boldsymbol{Q}, \boldsymbol{K}, \boldsymbol{V} \in \mathbb{R}^{n \times d}$ containing $d$-dimensional representations for $n$ queries, keys and values, the *dot-product self-attention* at a single head is computed

in the following way (Vaswani et al., 2017):

$$\boldsymbol{O} = \pi\left(\underbrace{\frac{\boldsymbol{Q}\boldsymbol{K}^\top}{\sqrt{d}}}_{\boldsymbol{S} \in \mathbb{R}^{n \times n}}\right)\boldsymbol{V} \in \mathbb{R}^{n \times d}. \quad (1)$$

The $\pi$ transformation usually maps rows to distributions, with $\pi(\boldsymbol{S})_{ij} = \text{softmax}\,(\boldsymbol{s}_i)_j$ being a common choice. For decoder-only models, $\boldsymbol{S}$ is masked in order to ignore the contribution from future tokens. Notably, a naive implementation of Equation 1 leads to a $\mathcal{O}\left(n^2\right)$ time and memory complexity for training.

### 2.3. FlashAttention

To address the costs of naive attention implementations, Dao et al. (2022) introduced FlashAttention, an algorithm that avoids the materialization of quadratic matrices via a GPU-aware implementation of online softmax (Milakov & Gimelshein, 2018), bringing the overall memory complexity to $\mathcal{O}\left(n\right)$. Subsequent versions of FlashAttention further improved GPU usage by reordering the loops, reducing the number of non-GEMM (general matrix multiply) operations (Dao, 2024), and exploiting the asynchronicity and support for FP8 low-precision on the new Hopper GPUs (Shah et al., 2024). The key idea of FlashAttention is to split the inputs $\boldsymbol{Q}, \boldsymbol{K}, \boldsymbol{V}$ into blocks, load them from slow GPU high bandwidth memory (HBM) to the fast GPU on-chip SRAM, then compute the attention output regarding those blocks and, at the end, scale the output by the right normalization factor.

### 2.4. Sparse Attention

The original softmax-based attention is *dense*, i.e., it puts *some* probability mass on all tokens—not only a computational disadvantage, but also making interpretation and generalization harder (Voita et al., 2019; Treviso et al., 2022; Veličković et al., 2025). An alternative to softmax is the **$\alpha$-entmax transformation** (Peters et al., 2019), which is differentiable and leads to sparse outputs:

$$\alpha\text{-entmax}\,(\boldsymbol{s}) = [(\alpha - 1)\boldsymbol{s} - \tau\mathbf{1}]_+^{1/\alpha - 1}, \quad (2)$$

where $[\cdot]_+$ is the ReLU function, and $\tau \in \mathbb{R}$ is a normalizing constant to ensure the output is a valid probability distribution. Importantly, entries with score $s_i \leq \frac{\tau}{\alpha - 1}$ get exactly zero probability. In the limit $\alpha \to 1$, $\alpha$-entmax recovers the softmax function, while for any value of $\alpha > 1$ this transformation returns increasingly sparser probability vectors. When $\alpha = 2$, we recover the sparsemax transformation (Martins & Astudillo, 2016). However, in contrast to fixed sparse patterns, such as windowed sparse attention (Child et al., 2019; Beltagy et al., 2020) and block-sparse variants (Zaheer et al., 2020b; Dao et al., 2022), $\alpha$-entmax's sparsity patterns are dynamic and hence difficult to exploit

in order to reduce the quadratic burden of self-attention because we still need to materialize $\boldsymbol{S} = \boldsymbol{Q}\boldsymbol{K}^\top$ before applying the transformation.

In the next section (§3), we outline ADASPLASH, our new method for computing $\alpha$-entmax attention, along with a novel custom Triton kernel (Tillet et al., 2019) that enables efficient training of transformers for extremely long context lengths. As shown in §4, our implementation maintains competitiveness with state-of-the-art algorithms such as FlashAttention by leveraging the sparsity given by $\alpha$-entmax, effectively exploiting the advantages of sparse attention at scale.

## 3. ADASPLASH

We start by revisiting the computation of $\alpha$-entmax for general values of $\alpha$ in §3.1, and proposing a new algorithm that has a fast empirical convergence. We design an efficient Triton kernel in §3.2, dubbed ADASPLASH, that effectively leverages adaptive sparsity patterns in both the forward and backward passes of $\alpha$-entmax in order to minimize runtime.

### 3.1. $\alpha$-entmax Computation

In order to compute Equation 2 for a given $\boldsymbol{s} \in \mathbb{R}^n$, we need to find the threshold $\tau \in \mathbb{R}$ such that the resulting output sums to 1. Mathematically, this is equivalent to finding the root of the following equation:

$$f(\tau) = \sum_i \left[(\alpha - 1)s_i - \tau\right]_+^{1/(\alpha-1)} - 1. \qquad (3)$$

**Exact algorithms for $\alpha \in \{1.5, 2\}$.** In particular, for $\alpha = 2$, the computation is reduced to an Euclidean projection onto the probability simplex, for which efficient algorithms have been extensively studied (Held et al., 1974; Duchi et al., 2008; Condat, 2016). Similarly, for $\alpha = 1.5$, Peters et al. (2019) introduced an exact sort-based algorithm. However, these methods either require complex data structures that are not efficiently handled in GPUs, or sorting-based algorithms, which require the materialization of the entire input.

**Bisection algorithm for $\alpha > 1$.** For a general $\alpha$, Blondel et al. (2019) introduced a bisection update rule to approximate $\tau$ by iteratively refining its lower ($\tau_{\text{lo}}$) and higher ($\tau_{\text{hi}}$) bounds:

$$B_f(\tau) = \begin{cases} (\tau_{\text{lo}}, \tau) & \text{if } f(\tau) < 0, \\ (\tau, \tau_{\text{hi}}) & \text{otherwise,} \end{cases} \qquad (4)$$

obtaining $\tau = \frac{1}{2}(\tau_{\text{lo}} + \tau_{\text{hi}})$ after the last iteration. While the bisection algorithm is simple and effective, it converges at a linear rate (Kaufman & Lenker, 1986), meaning the absolute error decreases by approximately half at each iter-

---

**Algorithm 1** Halley-bisection algorithm for $\alpha$-entmax.

1: **Input:** logits $\boldsymbol{s} \in \mathbb{R}^n$, param. $\alpha \in \mathbb{R}$, iterations $T$
2: Define $f(\tau) := \sum_i [s_i - \tau]_+^{1/(\alpha-1)} - 1$
3: Set $\boldsymbol{s} \leftarrow (\alpha - 1)\boldsymbol{s}$
4: Initialize $\tau_{\text{lo}} = \max(\boldsymbol{s}) - 1$
5: Initialize $\tau_{\text{hi}} = \max(\boldsymbol{s}) - n^{1-\alpha}$
6: Initialize $\tau = (\tau_{\text{lo}} + \tau_{\text{hi}})/2$
7: **repeat**
8:     Compute $\tau_{\text{lo}}, \tau_{\text{hi}} = B_f(\tau)$ (Equation 4)
9:     Compute $\tau_H = H_f(\tau)$ (Equation 5)
10:    **if** $\tau_H \in [\tau_{\text{lo}}, \tau_{\text{hi}}]$ **then**
11:       $\tau \leftarrow \tau_H$         ▷ (Halley's Update)
12:    **else**
13:       $\tau \leftarrow \frac{1}{2}(\tau_{\text{lo}} + \tau_{\text{hi}})$    ▷ (Bisection Update)
14:    **end if**
15: **until** $T$ iterations are completed
16: **Output:** $[\boldsymbol{s} - \tau\mathbf{1}]_+^{1/(\alpha-1)}$

---

ation. Achieving high precision often requires many iterations, resulting in frequent memory accesses. As a result, in memory-bound scenarios where the time taken is mostly determined by the number of memory accesses—such as in attention—the number of iterations can significantly impact the runtime cost.

**Halley-bisection algorithm.** In order to obtain a faster runtime, we propose a hybrid algorithm for solving Equation 3 for any $\alpha > 1$ that combines the convergence guarantee of bisection with the faster convergence of Halley's method (Scavo & Thoo, 1995). As we show in §4.1, this approach achieves significant wall-clock speed-ups while requiring fewer iterations to attain the same precision.

The function defined in Equation 2 enjoys a cheap computation of its derivatives. Thus, methods that incorporate second-order information, such as Halley's method, can be leveraged to improve the approximation of $\tau$ at each iteration. Halley's method, which uses both the first and second derivatives, updates the solution using the following rule:

$$H_f(\tau) = \tau - \frac{2f(\tau)f'(\tau)}{2f'(\tau)^2 - f(\tau)f''(\tau)}, \qquad (5)$$

where the derivatives are given as follows:

$$f'(\tau) = -\frac{1}{\alpha - 1} \sum_i \left[(\alpha - 1)s_i - \tau\right]_+^{1/(\alpha-1)-1}, \qquad (6)$$

$$f''(\tau) = \frac{2 - \alpha}{(\alpha - 1)^2} \sum_i \left[(\alpha - 1)s_i - \tau\right]_+^{1/(\alpha-1)-2}. \qquad (7)$$

While Halley's method offers faster convergence under ideal conditions, it does not always converge, particularly when

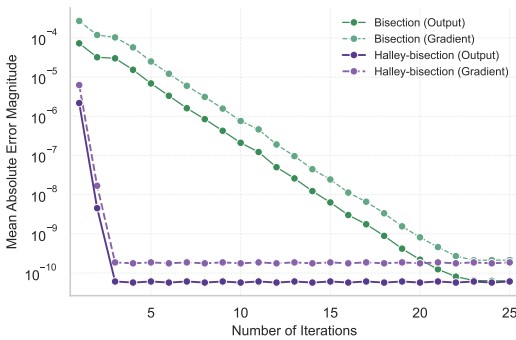

*Figure 2.* Comparison of mean absolute error magnitudes between Halley-bisection and Torch's bisection methods across iterations, measured against the exact solution for $\alpha = 1.5$.

the initial guess is far from the solution. To ensure convergence, we introduce a fail-safe mechanism that integrates the convergence guarantee of bisection: whenever Halley's method produces an update that moves the solution out of the bisection bounds, the algorithm reverts to a bisection update $B_f(\tau)$. This ensures that the algorithm converges, even in the worst cases, while leveraging the cubic convergence of Halley's method wherever possible. We outline our hybrid algorithm in Algorithm 1.

**Efficiency Benchmark.** We compare the runtime of Halley-bisection against existing algorithms for computing $\alpha$-entmax implemented in Torch. Specifically, we generate random tensors from a standard Gaussian distribution ($\mu = 0, \sigma^2 = 1$) with a fixed sequence length of $n = 8192$. For each configuration, we measure the average runtime over 1000 runs. Overall, we observe that Halley-bisection is significantly more efficient than the standard bisection algorithm implemented in Torch. Halley-bisection achieves a runtime of 2.38 ms, compared to 36.67 ms for the standard bisection algorithm, making it approximately **15× faster**. In addition, Halley-bisection reduces memory usage by **1.75×**, requiring only 512 MB compared to 896.15 MB for bisection. Furthermore, in Figure 2 we show that Halley-bisection ($\alpha = 1.5$) requires only 3 iterations to converge to machine precision for both the output and the gradient. On the other hand, the standard bisection algorithm takes 23 iterations to achieve the same precision for both cases.

### 3.2. Flash $\alpha$-entmax Attention

Given an algorithm to compute the entmax mapping that requires $T$ iteration steps, a naive implementation of entmax attention proceeds as follows: (1) multiply $\boldsymbol{S} = \boldsymbol{Q}\boldsymbol{K}^\top \in \mathbb{R}^{n \times n}$ and write the result to slow HBM on the GPU; (2) load $\boldsymbol{S}$ from HBM $T$ times to compute $\boldsymbol{\tau}$; (3) load $\boldsymbol{S}$ from HBM again, and write the result $\boldsymbol{P} = \alpha\text{-entmax}(\boldsymbol{S})$ to HBM; (4) perform a matrix multiplication to get the output

---

**Algorithm 2** ADASPLASH forward pass (w/o masking)

1: **Require:** Matrices $\boldsymbol{Q}, \boldsymbol{K}, \boldsymbol{V} \in \mathbb{R}^{n \times d}$ in HBM, block sizes $B_c, B_r$, param. $\alpha \in \mathbb{R}$
2: Divide $\boldsymbol{Q}$ into $T_r = \lceil n/B_r \rceil$ blocks $\boldsymbol{Q}_1, \ldots, \boldsymbol{Q}_{T_r}$ of size $B_r \times d$
3: Divide $\boldsymbol{K}, \boldsymbol{V}$ into $T_c = \lceil n/B_c \rceil$ blocks $\boldsymbol{K}_1, \ldots, \boldsymbol{K}_{T_c}$, $\boldsymbol{V}_1, \ldots, \boldsymbol{V}_{T_c}$ of size $B_c \times d$
4: Divide $\boldsymbol{O} \in \mathbb{R}^{n \times d}$ into $T_r$ blocks $\boldsymbol{O}_1, \ldots, \boldsymbol{O}_{T_r}$ of size $B_r \times d$
5: Divide $\boldsymbol{\tau}$ into $T_r$ blocks $\boldsymbol{\tau}_1, \ldots, \boldsymbol{\tau}_{T_r}$ of size $B_r$
6: **for** $i = 1$ to $T_r$ **do**
7: $\quad$ Load $\boldsymbol{Q}_i$ from HBM to on-chip SRAM
8: $\quad$ On chip, initialize $\boldsymbol{O}_i$
9: $\quad$ On chip, compute $\boldsymbol{\tau}_i$ using Hybrid Halley's with predefined $\alpha$, using a block version of Algorithm 1.
10: $\quad$ **for** $j = 1$ to $T_c$ **do**
11: $\quad\quad$ Load $\boldsymbol{K}_j, \boldsymbol{V}_j$ from HBM to on-chip SRAM
12: $\quad\quad$ Compute $\boldsymbol{S}_i^{(j)} = \boldsymbol{Q}_i\boldsymbol{K}_j^\top \in \mathbb{R}^{B_r \times B_c}$
13: $\quad\quad$ Compute $\boldsymbol{P}_i^{(j)} = \left[(\alpha - 1)\boldsymbol{S}_i^{(j)} - \boldsymbol{\tau}_i\right]_+^{1/\alpha - 1}$
14: $\quad\quad$ Accumulate $\boldsymbol{O}_i \leftarrow \boldsymbol{O}_i + \boldsymbol{P}_i^{(j)}\boldsymbol{V}_j$
15: $\quad$ **end for**
16: $\quad$ Write $\boldsymbol{O}_i$ and $\boldsymbol{\tau}_i$ to HBM
17: **end for**
18: **Return:** Output $\boldsymbol{O}$ and $\boldsymbol{\tau}$

---

$\boldsymbol{O} = \boldsymbol{PV}$. However, since most of these operations are memory-bound, the excessive number of HBM accesses leads to slow wall-clock times. Moreover, having to materialize $\boldsymbol{S}$ and $\boldsymbol{P}$ in memory poses a major bottleneck, as their sizes quickly exceed GPU memory capacity when the sequence length $n$ increases. To address these issues and speed up $\alpha$-entmax attention on hardware accelerators like GPUs, we propose an algorithm that reduces HBM reads and writes while producing the same outputs as the naive implementation.

#### 3.2.1. FORWARD PASS

We outline the forward pass in Algorithm 2 (without masking full-zero blocks, which we introduce later on this section). Concretely, given the inputs $\boldsymbol{Q}, \boldsymbol{K}, \boldsymbol{V} \in \mathbb{R}^{n \times d}$ stored in HBM, the goal is to compute the attention output $\boldsymbol{O} \in \mathbb{R}^{n \times d}$ efficiently and write it back to HBM. Akin to the approach taken in FlashAttention (Dao et al., 2022), we employ two well-known techniques—**tiling** and **recomputation**—to address the challenge of materializing the matrices $\boldsymbol{S} \in \mathbb{R}^{n \times n}$ and $\boldsymbol{P} \in \mathbb{R}^{n \times n}$.

**Tiling.** The key idea involves splitting the inputs $\boldsymbol{Q}, \boldsymbol{K}, \boldsymbol{V}$ into smaller blocks, and then computing attention block by block. We start by loading only $\boldsymbol{Q}$ and $\boldsymbol{K}$ from the slower HBM to the faster SRAM to compute $\boldsymbol{\tau} \in \mathbb{R}^n$ using the

Halley-bisection algorithm (Alg. 1). In order to use the aforementioned algorithm, we need to accumulate three values: $f(\boldsymbol{\tau})$, $f'(\boldsymbol{\tau})$, $f''(\boldsymbol{\tau})$. Since $f$, as well as its derivatives, is additive over its inputs, their computation can also be computed in blocks. Let $B_r$ and $B_c$ be the row and column block sizes, respectively, and define $T_r = \lceil n/B_r \rceil$ and $T_c = \lceil n/B_c \rceil$. Divide $\boldsymbol{Q}$ into $\boldsymbol{Q}_1, ..., \boldsymbol{Q}_{T_r}$ blocks, and $\boldsymbol{K}$ into $\boldsymbol{K}_1, ..., \boldsymbol{K}_{T_c}$ blocks. Then, $f(\boldsymbol{\tau})$ can be computed as:

$$f(\boldsymbol{\tau}_i) = \sum_{j=1}^{T_c} f(\boldsymbol{\tau}_i; \boldsymbol{S}_i^{(j)}) \qquad (8)$$

where $\boldsymbol{S}_i^{(j)} = \boldsymbol{Q}_i \boldsymbol{K}_j^\top \in \mathbb{R}^{B_r \times B_c}$ and $\boldsymbol{\tau}_i$ represents the $i^{\text{th}}$ sliced block of $\boldsymbol{\tau}$ with size $T_r$. Thus, these quantities do not need to ever be materialized and can be accumulated directly in fast memory. Afterwards, we load $\boldsymbol{V}$ to compute the attention output $\boldsymbol{O}$ for those blocks. In contrast to FlashAttention, our approach requires loading $\boldsymbol{K}$ to compute $\boldsymbol{S}$ at least two additional times. Therefore, the forward pass is bound to always be slower than FlashAttention's due to the extra HBM reads and computation.

**Recomputation.** In order to avoid the materialization of the matrices $\boldsymbol{S}$ and $\boldsymbol{P}$, we recompute them again in Algorithm 1, which is used to compute $\boldsymbol{\tau}$, and also recompute them for obtaining the gradients for the backward pass. By doing this we are increasing the required FLOPs to reduce the maximum amount of memory required. While this might suggest an increase in runtime, the opposite is observed (Dao et al., 2022). Despite the need for additional matrix multiplications, the reduction in total HBM reads and writes more than offsets the extra FLOPs, leading to improved performance overall.

**Sparsity-aware implementation.** The key challenge of $\alpha$-entmax attention lies in finding the threshold $\boldsymbol{\tau}$, which requires multiple evaluations of the function $f(\boldsymbol{\tau})$, which, in turn, depends on the score matrix $\boldsymbol{S}$. While our proposed Halley-bisection algorithm alleviates the number of iterations needed to recompute $\boldsymbol{S}_i^{(j)}$ by providing a faster empirical convergence, our current implementation still iterates over all blocks of $\boldsymbol{S}$, including **null blocks**—blocks where the corresponding entries of the sparse attention matrix $\boldsymbol{P}$ are zero.

Furthermore, empirical evidence from Jiang et al. (2024) and (Xiao et al., 2024) suggests that for long inputs (e.g., 128k tokens in LLaMa-3-8b), approximately 3% of the entries in $\boldsymbol{P}$ suffice to capture over 96% of the total attention, which motivates an approach to leverage the adaptive and unstructured sparsity of $\alpha$-entmax attention weights. To this end, we propose to only compute necessary blocks of $\boldsymbol{P}$ by skipping the null blocks. Concretely, let $\mathcal{I}(i)$ denote the set of all indices $i'$ such that $\lfloor i'/T_r \rfloor = i$, and $\mathcal{J}(j)$ denote the

set of all indices $j'$ such that $\lfloor j'/T_c \rfloor = j$. We construct a **block mask** matrix $\boldsymbol{M} \in \{0,1\}^{T_r \times T_c}$ as follows:

$$M_{ij} = \begin{cases} 1 & \text{if } \exists_{i' \in \mathcal{I}(i), j' \in \mathcal{J}(j)} : S_{i',j'} > \tau_{i'}, \\ 0 & \text{otherwise,} \end{cases} \qquad (9)$$

Importantly, $\boldsymbol{M}$ is created dynamically after a small predefined number of Halley-bisection iterations.

While the introduction of $\boldsymbol{M}$ breaks the linear memory complexity of dense fused-attention by requiring $T_r \times T_c$ extra memory, the overhead is still manageable as it only contains binary values and is substantially smaller than the full $\boldsymbol{P} \in \mathbb{R}^{n \times n}$ matrix. Furthermore, $\boldsymbol{M}$ needs to be materialized only once and its memory can be shared across all attention layers. To leverage $\boldsymbol{M}$ in practice, we propose to create two **pointer-increment lookup tables**:

1. $\mathcal{K}_j = \{i \mid M_{ij} = 1\}$: A table containing the row indices $i$ of $\boldsymbol{M}$ that lead to non-null blocks in $\boldsymbol{P}_i^{(j)}$.

2. $\mathcal{Q}_i = \{j \mid M_{ij} = 1\}$: A table containing the column indices $j$ of $\boldsymbol{M}$ that lead to non-null blocks in $\boldsymbol{P}_i^{(j)}$.

These tables enable efficient skipping of $\boldsymbol{K}$ and $\boldsymbol{V}$ blocks that do not contribute to the final attention output $\boldsymbol{O}$, significantly reducing unnecessary computations. Moreover, the same mechanism can be extended to accelerate the backward pass, where gradients with respect to $\boldsymbol{Q}$, $\boldsymbol{K}$, and $\boldsymbol{V}$ are computed, which we describe next.

### 3.2.2. BACKWARD PASS

In FlashAttention (Dao et al., 2022), the backward pass is executed using a single kernel that parallelizes computation across batch, head, and sequence dimensions. However, following Triton's official implementation of FlashAttention,[2] we separate the backward pass into two kernels: one for $\boldsymbol{dQ}$ (the gradient w.r.t. $\boldsymbol{Q}$) and another for $\boldsymbol{dK}$ and $\boldsymbol{dV}$ (the gradients w.r.t. $\boldsymbol{K}$ and $\boldsymbol{V}$).

**Sparse Jacobian of $\alpha$-entmax.** The sparsity in the Jacobian of $\alpha$-entmax plays a crucial role in the backward pass. For $\boldsymbol{p} = \alpha$-entmax$(\boldsymbol{s})$, the Jacobian is (Peters et al., 2019)

$$\frac{\partial \alpha\text{-entmax}(\boldsymbol{s})}{\partial \boldsymbol{s}} = \text{Diag}(\boldsymbol{u}) - \frac{\boldsymbol{u}\boldsymbol{u}^\top}{\|\boldsymbol{u}\|_1}, \qquad (10)$$

where $u_j = (p_j)^{2-\alpha}$. Importantly, this Jacobian is sparse and only depends on $\boldsymbol{p}$, which, in turn, is a function of $\tau$ computed during the forward pass. We denote by $\boldsymbol{U} \in \mathbb{R}^{n \times n}$ the matrix defined element-wise as $U_{lk} = P_{lk}^{2-\alpha}$, and by $\boldsymbol{U}_i^{(j)} \in \mathbb{R}^{B_r \times B_c}$ its $(i, j)^{\text{th}}$ block. Using this information,

---

[2] https://github.com/triton-lang/triton/blob/main/python/tutorials/06-fused-attention.py

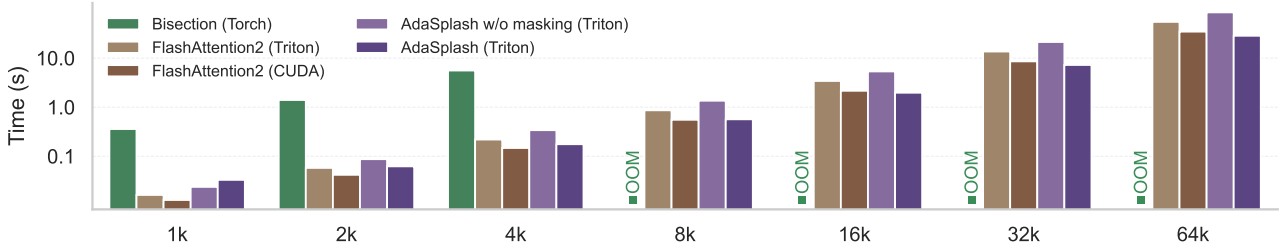

*Figure 3.* Efficiency of algorithms for computing non-causal attention in terms of the average training step time for increasingly longer sequence lengths. We use $\alpha = 1.5$ for $\alpha$-entmax based methods (Bisection and ADASPLASH).

the gradient w.r.t. the score matrix $\boldsymbol{S}_i^{(j)} \in \mathbb{R}^{B_r \times B_c}$ can be efficiently computed as:

$$d\boldsymbol{S}_i^{(j)} = \boldsymbol{U}_i^{(j)} \odot d\boldsymbol{P}_i^{(j)} - \text{Diag}(\boldsymbol{\delta}_i)\boldsymbol{U}_i^{(j)}, \quad (11)$$

where $d\boldsymbol{P}_i^{(j)} = d\boldsymbol{O}_i \boldsymbol{V}_j^\top \in \mathbb{R}^{B_r \times B_c}$, with $d\boldsymbol{O}_i \in \mathbb{R}^{B_r \times n}$ and $\boldsymbol{V}_j \in \mathbb{R}^{B_c \times n}$, and $\boldsymbol{\delta}_i \in \mathbb{R}^{B_r}$ denotes the $i^{\text{th}}$ block of the vector $\boldsymbol{\delta} \in \mathbb{R}^n$ defined element-wise as $\delta_l = (\sum_k U_{lk} dP_{lk})/(\sum_k U_{lk})$.

**Efficient gradient computation.** In ADASPLASH, instead of storing $\boldsymbol{P}$, we store the lookup tables $\mathcal{K}$ and $\mathcal{Q}$ computed during the forward pass, allowing us to to efficiently skip the computations of null blocks during backpropagation. Given $d\boldsymbol{S}_i$, the gradients for $\boldsymbol{Q}_i, \boldsymbol{K}_i, \boldsymbol{V}_i \in \mathbb{R}^{B_r \times d}$ are computed as follows using the pointer-increment lookup tables:

$$d\boldsymbol{Q}_i = \sum_{j \in \mathcal{Q}_i} d\boldsymbol{S}_i^{(j)} \cdot \boldsymbol{K}_j, \quad (12)$$

$$d\boldsymbol{K}_j = \sum_{i \in \mathcal{K}_j} d\boldsymbol{S}_i^{(j)} \cdot \boldsymbol{Q}_i, \quad (13)$$

$$d\boldsymbol{V}_j = \sum_{i \in \mathcal{K}_j} \boldsymbol{P}_i^{(j)} \cdot d\boldsymbol{O}_i. \quad (14)$$

Hence, by splitting the backward pass into separate kernels and exploiting the sparsity of $\alpha$-entmax through the Jacobian structure, we can achieve efficient gradient computation. Overall, ADASPLASH allows users to choose between memory efficiency (without block masking) and computational speed (with block masking) depending on the task requirements and hardware constraints. We provide a detailed derivation of $\alpha$-entmax attention's backward pass and its implementation in Appendix A.2.

## 4. Experiments

We evaluate ADASPLASH across various scenarios to show its computational efficiency and impact on downstream tasks. Our experiments address the following questions:

- Performance efficiency: How does ADASPLASH compare with baseline methods in terms of runtime as sequence length and sparsity vary?

- Generalization to architectures: How does ADAS-PLASH perform when integrated with encoder-only and decoder-only models?

- Effectiveness in finetuning: Can ADASPLASH-pretrained models outperform or match their dense counterparts in short and long-context tasks?

### 4.1. Efficiency Benchmark

We compare the efficiency of ADASPLASH against FlashAttention-2 and naive implementations of $\alpha$-entmax. For a fair comparison, we also include a variant of FlashAttention-2 implemented in Triton that follows closely our kernel implementation of ADASPLASH. We set the number of iterations of ADASPLASH to 3 and Bisection to 10. As input, we generate random tensors from a Gaussian distribution ($\mu = 0$), simulating attention scores with a high level of sparsity by setting the Gaussian variance to $\sigma^2 = 6$ of query vectors. Sequence lengths range from 1k to 64k, with a fixed head size of $d = 64$.

**Runtime.** We show the average training step time for each method in Figure 3. ADASPLASH demonstrates superior scalability, efficiently handling sequences up to 64k, unlike the Bisection method implemented in Torch, which runs out of memory beyond 4k context length. We also note that, as context length increases, the amount of block sparsity naturally increases as well, leading to an advantage for our method over both implementations of FlashAttention-2.

### 4.2. Performance on Real Tasks

Encoder-only models, such as RoBERTa (Liu et al., 2019) and ModernBERT (Warner et al., 2024), exhibit higher attention sparsity than decoder-only models, making them well-suited for adaptive sparse attention mechanisms like ADASPLASH. Following ModernBERT's evaluation setup, we opt to evaluate these models on standard NLP tasks, such as text classification, natural language inference, textual similarity, and information retrieval. Moreover, following FlashAttention's evaluation setup (Dao et al., 2022), we also benchmark ADASPLASH with GPT-2, a decoder-only

*Table 1.* Results for single-vector retrieval models on different tasks from the BEIR benchmark in terms of nDCG@10.

| Model | Seq. | SciFact | NFC | FiQA | TREC-C |
|---|---|---|---|---|---|
| RoBERTa | 512 | 51.7 | 23.1 | 27.8 | 60.1 |
| RoBERTa ($\alpha = 1.5$) | 512 | 50.8 | 24.2 | 27.6 | 71.0 |
| RoBERTa ($\alpha = 2.0$) | 512 | 52.2 | 23.8 | 25.7 | 65.5 |
| ModernBERT | 8192 | 57.7 | 22.4 | 25.7 | 67.6 |
| ModernBERT ($\alpha = 1.5$) | 8192 | **58.4** | **25.7** | **29.6** | **75.2** |
| ModernBERT ($\alpha = 2.0$) | 8192 | 58.0 | 25.4 | 29.3 | 71.1 |

*Table 2.* Long document classification performance ($F_1$ micro) with softmax and $\alpha$-entmax attention.

| | Sequence Length | | | | |
|---|---|---|---|---|---|
| Model | 512 | 1024 | 2048 | 4096 | 8192 |
| RoBERTa | 71.5 | 74.4 | 75.1 | 77.9 | **79.2** |
| RoBERTa ($\alpha = 1.5$) | **71.8** | **75.5** | **76.4** | **78.0** | 78.6 |

model, to assess its efficiency in autoregressive settings where attention patterns are denser. This ensures a comprehensive comparison with optimized softmax-based methods while validating the benefits of sparsity across different architectures. We provide more training and evaluation details for each task in Appendix B.

**Continuous pretraining.** We conducted continuous pretraining of RoBERTa-base and ModernBERT-base on 2B tokens of the English subset of Fineweb-edu (Lozhkov et al., 2024) using ADASPLASH for $\alpha \in \{1.5, 2\}$, and PyTorch's `scaled_dot_product_attention` for $\alpha = 1.0$. To ensure a smooth transition from dense to sparse attention, we linearly increased $\alpha$ from $\alpha = 1.0$ to the target values $\alpha \in \{1.5, 2.0\}$ over the first 1B tokens and kept it fixed afterwards. We provide more details on the continuous pretraining phase in Appendix B.1, including efficiency results.

**Single-vector retrieval.** We evaluate our pretrained models on single-vector retrieval performance using the BEIR benchmark (SciFact, NFCorpus, FiQA2018, TREC-COVID), following the setup in (Warner et al., 2024). Table 1 highlights the performance of RoBERTa and ModernBERT models using $\alpha$-entmax attention in terms of the standard nDCG@10 metric. ModernBERT with $\alpha = 1.5$ consistently outperformed its dense counterpart, achieving the highest scores on all tasks, demonstrating its ability to focus on relevant signals effectively. While ModernBERT with $\alpha = 2.0$ remained competitive, its higher sparsity might have excluded relevant information, affecting task performance. Finally, sparse versions of ModernBERT achieve better results than the sparse versions of RoBERTa on all tasks, highlighting the benefit of modeling long contexts.

*Table 3.* Runtime per epoch (hh:mm:ss) and peak memory usage (GB) for long document classification with different sequence lengths. In cases where the full batch could not fit in memory, gradient accumulation was used. Memory values represent the effective peak memory required to process a batch of 16 samples.

| Runtime (hh:mm:ss) | Sequence Length | | | | |
|---|---|---|---|---|---|
| Model | 512 | 1024 | 2048 | 4096 | 8192 |
| RoBERTa | 2:39 | 5:00 | 9:35 | 18:36 | 35:51 |
| RoBERTa ($\alpha = 1.5$) | 2:45 | 5:20 | 10:24 | 19:54 | 38:08 |
| w/ Torch Bisect | 4:51 | 8:44 | 22:48 | 1:11:53 | 4:12:34 |

| Memory (GB) | Sequence Length | | | | |
|---|---|---|---|---|---|
| | 512 | 1024 | 2048 | 4096 | 8192 |
| RoBERTa | 6.75 | 11.43 | 20.35 | 37.49 | 75.00 |
| RoBERTa ($\alpha = 1.5$) | 6.75 | 11.45 | 20.38 | 39.17 | 79.88 |
| w/ Torch Bisect | 7.75 | 16.92 | 44.06 | 142.76 | 508.16 |

**Long document classification.** We fine-tuned a pretrained RoBERTa model (Liu et al., 2019) on the ECtHR (Chalkidis et al., 2019; 2021) dataset while progressively increasing the sequence length up to 8192 tokens. Positional embeddings were extended by repetition, following the approach of Beltagy et al. (2020). As a baseline, we fine-tuned the model using standard softmax-based attention. For $\alpha$-entmax attention, we linearly increased the $\alpha$ from 1.0 to 1.5 during training to ensure smooth convergence. The results, summarized in Table 2, show a consistent improvement in model performance with longer context lengths. Notably, despite the base model being pretrained with standard attention, $\alpha$-entmax attention was capable of effectively learning the task, achieving a slightly higher micro $F_1$ score than the model fine-tuned with standard attention up to a sequence length of 4096 tokens.

Table 3 compares the runtime per epoch and peak memory usage for different sequence lengths on the long document classification task. We report results for RoBERTa with FlashAttention-2 ($\alpha = 1$), RoBERTa with ADASPLASH ($\alpha = 1.5$), and RoBERTa using Torch's bisection-based implementation. ADASPLASH enables scalable training with $\alpha$-entmax attention. Prior to this, implementations had to resort to Torch's bisection, which leads to both extremely slow runtimes or even out-of-memory problems, rendering it infeasible for most realistic training setups. In contrast, our method brings the cost of $\alpha$-entmax attention down to the level of existing dense attention implementations, as both runtime and memory usage with ADASPLASH remain well aligned with those of FlashAttention-2.

**Language understanding.** We also evaluate RoBERTa and ModernBERT models with $\alpha$-entmax attention on the GLUE benchmark (Wang et al., 2018) in Appendix B.2. Overall, the results indicate that models with sparse attention

*Table 4.* Results on language modeling with GPT-2 in terms of final validation loss and accuracy on the HellaSwag task (Zellers et al., 2019), along with the average runtime per training step (in seconds) and peak memory usage (GB) per GPU.

| Model | Val. Loss | HS Acc. | Runtime | Memory |
|---|---|---|---|---|
| GPT-2 | 3.283 | 30.4 | **0.98** | **52.5** |
| GPT-2 ($\alpha = 1.5$) | **3.263** | **30.6** | 1.03 | **52.5** |
| w/ Torch sorting | - | - | 3.61 | 73.8 |
| w/ Torch bisection | - | - | 7.78 | 77.6 |

achieve comparable performance to their dense counterparts, which underscores the ability to efficiently train $\alpha$-entmax models without sacrificing accuracy.

**Language modeling.** Following (Dao et al., 2022), we trained a small 124M GPT-2 model (Radford et al., 2019) from scratch on 10B tokens of the FineWeb dataset (Penedo et al., 2024) with a context length of 1024 tokens. For a consistent evaluation between softmax and $\alpha$-entmax attention, we also trained a softmax-based GPT-2 to serve as baseline. After training, we evaluated both models on the HellaSwag task (Zellers et al., 2019). Table 4 presents a side-by-side comparison of the final validation loss and accuracy on HellaSwag, along with runtime and memory usage numbers. Sparse GPT-2 achieves a slight improvement in validation loss (3.263 vs. 3.283) and final accuracy (30.6% vs. 30.4%) compared to its softmax counterpart, while obtaining comparable runtime and memory efforts. Furthermore, our approach achieves a runtime comparable to the GPT-2 using the highly optimized FA2 (1.03 s/step vs. 0.98 s/step) and matches its memory footprint (52.5 GB), while outperforming the sorting and bisection variants by large margins in both speed (1.03 s/step vs. 3.61 and 7.78 s/step) and memory usage (52.5 GB vs. 73.8 and 77.6 GB). In Appendix B.4, we report all training and evaluation details, including the validation loss curves of each method.

**Sparsity in attention heads.** Figure 4 presents the sparsity observed in attention heads for all layers for an input of 1024 tokens for our sparse GPT-2 model ($\alpha = 1.5$). Except for the first layer, all subsequent layers exhibit a high degree of sparsity, highlighting the potential efficiency gains from leveraging this property. Moreover, in Figure 5 (Appendix B.1), we illustrate the sparsity patterns in ModernBERT-base attention heads for $\alpha \in \{1.5, 2.0\}$, reinforcing similar conclusions.

## 5. Related Works

**Sparse Probability Transformations.** The sparsity inherent to the $\alpha$-entmax transformation, as demonstrated by Blondel et al. (2019), is directly controlled by the $\alpha$ pa-

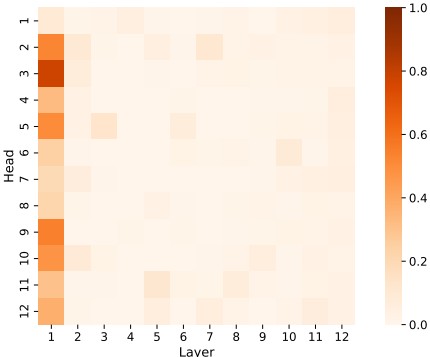

*Figure 4.* Ratio of non-zero attention scores for GPT-2 ($\alpha = 1.5$).

rameter. For $\alpha = 2$, the problem simplifies to a projection onto the probability simplex, a well-established optimization problem. Its solution forms the base of sparsemax (Martins & Astudillo, 2016), which can be efficiently computed using sorting and root-finding methods (Held et al., 1974; Condat, 2016; Liu & Ye, 2009). Moreover, for intermediate values such as $\alpha = 1.5$, Peters et al. (2019) proposed an exact sorting-based algorithm along with an implementation of a bisection algorithm applicable to any $\alpha$. However, these approaches remain suboptimal for long contexts due to slow convergence or reliance on complex data structures and sorting operations, which are difficult to optimize for hardware.

**Sparse Attention Mechanisms.** Efficient sparse attention mechanisms have been widely studied to reduce the quadratic cost of transformers. The Sparse Transformer (Child et al., 2019) introduces a fixed windowed attention that can be efficiently computed using CUDA kernels, a strategy also adopted by Longformer (Beltagy et al., 2020), and BigBird (Zaheer et al., 2020a). However, data-dependent sparse attention methods, such as Reformer (Kitaev et al., 2020) and Routing Transformer (Roy et al., 2021), aimi to approximate softmax in return for efficiency, not leveraging the sparsity of attention weights. Other methods, such as Top-$k$ attention (Gupta et al., 2021) and NSA (Yuan et al., 2025), provide sparsity but require a fixed, non-adaptable budget. In contrast, $\alpha$-entmax attention provides natural, input-dependent sparsity patterns with an exact and differentiable transformation that generalizes softmax, making it more flexible for modeling attention distributions. Adaptively sparse transformers (Correia et al., 2019) uses $\alpha$-entmax attention where attention heads can learn $\alpha$ dynamically, improving interpretability but without leveraging sparsity for efficiency. SparseFinder (Treviso et al., 2022) aims to address efficiency issues by predicting the sparsity pattern of entmax attention a priori; however, it does not scale efficiently for long sequences.

**Hardware-Aware Attention.** Recent works have explored optimizing attention mechanisms with hardware-aware implementations. Flex Attention (Dong et al., 2024) provides an API for efficient attention computation, though they remain tied to softmax-based transformations and do not support more complex operations such as those considered in our work. Closely related to our approach, FlashAttention-1 and 2 (Dao et al., 2022; Dao, 2024) optimize softmax-based attention using tiling and recomputation techniques implemented in CUDA. While FlashAttention includes a sparse block variant, its sparsity pattern must be predefined, limiting adaptability. In this work, we compare our method, ADASPLASH, with FlashAttention-2 and demonstrate that our approach can outperform both its CUDA and Triton implementations at high input sparsity levels. Similarly, Sparse Flash Attention (Pagliardini et al., 2023) extends FlashAttention-1 with a sparse variant that reduces computational cost by either dropping queries and keys per head or grouping them using a hash-based bucketing approach. However, despite its efficiency improvements, it relies on slow sorting operations and is constrained to causal attention, making its sparsity a by-product of bucketing rather than an inherently adaptive feature, as in our case.

**Efficiency at Inference Time.** Another line of work focuses on optimizing transformers at inference time. Methods such as Paged Attention (Kwon et al., 2023) and KV cache sparsification (Devoto et al., 2024; Luohe et al., 2024) aim to alleviate the linear complexity of inference by modifying key-value caching strategies. While our approach does not directly provide KV cache compression benefits, these methods are orthogonal and can be combined with our work to further improve inference efficiency.

## 6. Conclusion

In this work, we introduced ADASPLASH, a hardware-aware and efficient implementation of $\alpha$-entmax attention, bridging the gap between adaptive sparse activations and efficient long-context modeling. Our approach leverages a hybrid Halley-bisection algorithm for faster empirical convergence and custom Triton kernels to exploit the inherent sparsity of $\alpha$-entmax. Our experiments show that ADASPLASH not only achieves substantial computational improvements over existing $\alpha$-entmax implementations, but can often match or even surpass the efficiency of highly optimized softmax-based attention algorithms like FlashAttention-2. Moreover, ADASPLASH enables long-context training while maintaining strong task performance across diverse benchmarks, such as language understanding, information retrieval, document classification, and language modeling. Overall, our work unlocks the viability of dynamically sparse attention mechanisms in large-scale training, which was previously hindered by computational inefficiencies.

## Impact Statement

Efficient attention mechanisms are crucial for scaling transformers to long-context tasks. Our work provides a practical implementation by making adaptive sparse attention efficient, overcoming previous computational limitations of $\alpha$-entmax. Therefore, the improved efficiency of ADASPLASH has potential applications in large-scale NLP, where sparsity can be leveraged to reduce computational costs. We do not foresee direct societal consequences from sparsity itself, but its integration into decision-making models may still reflect biases in training data. As such, we encourage careful evaluation when deploying sparse attention mechanisms in high-stakes applications, ensuring that efficiency gains do not come at the cost of fairness or transparency.

## Acknowledgments

We thank Vlad Niculae for his insightful and constructive comments throughout this work. We also thank the SARDINE Lab members for reviewing this paper and providing helpful feedback. This work was supported by the Portuguese Recovery and Resilience Plan through project C645008882-00000055 (Center for ResponsibleAI), by the EU's Horizon Europe Research and Innovation Actions (UTTER, contract 101070631), by the project DECOLLAGE (ERC-2022-CoG 101088763), and by FCT/MECI through national funds and when applicable co-funded EU funds under UID/50008: Instituto de Telecomunicações.

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

# A. Algorithm Details

We first derive a high-level view of the forward and backward passes of the entmax attention and then present the full algorithms for both mentioned versions. For consistency and ease of comparison, we follow the notation adopted by FlashAttention-1 (Dao et al., 2022).

## A.1. $\alpha$-entmax Attention Forward Pass

We recall that given the input sequences $Q, K, V \in \mathbb{R}^{n \times d}$, we want to compute the attention output $O \in \mathbb{R}^{n \times d}$ as follows:

$$S = QK^\top \in \mathbb{R}^{n \times n}, \ P = \alpha\text{-entmax}(S) \in \mathbb{R}^{n \times n}, \ O = PV \in \mathbb{R}^{n \times d}$$

Therefore all we need is the $\tau \in \mathbb{R}^n$ that solves Equation 2, for which we can use Algorithm 1. We note that we do not need to materialize $S$ as we only need to accumulate the derivatives of $f(\tau)$, defined in Equation 3. Once $\tau$ is computed, we can compute each row of $O$ as follows:

$$O_i = P_i V = \sum_j P_{ij} V_j = \sum_{j=1}^{n} \max\left(0, (\alpha - 1)Q_i^\top K_j - \tau_i\right)^{1/\alpha - 1} V_j \tag{15}$$

As in FlashAttention, we can compute $O_i$ without extra memory by incrementally summing the contributions of each $\alpha$-entmax$(Q_i^\top K_j)V_j$ term. We can then compute the forward pass with $\mathcal{O}(n)$ extra memory as follows:

1. Compute $\tau_i$ for all $1 \leq i \leq n$ according to Algorithm 1, which takes $\mathcal{O}(n)$ extra memory.

2. Compute $O_i$ for all $1 \leq i \leq n$ according to Equation 15 which takes $\mathcal{O}(n)$ extra memory.

## A.2. $\alpha$-entmax Attention Backward Pass

For the $\alpha$-entmax attention backward pass, we need to compute the gradients with respect to $V$, $K$, and $Q$. Let $\mathcal{L}$ be a scalar loss function, and $dO \in \mathbb{R}^{n \times d}$ denote $\frac{\partial \mathcal{L}}{\partial O}$. Our goal is to compute the input gradients $dV, dK, dQ \in \mathbb{R}^{n \times d}$.

### 1. Gradient of $V$

Using reverse-mode autodifferentiation, we first compute $dV$:

$$dV = P^\top dO, \tag{16}$$

where $P = \alpha\text{-entmax}(S)$ is the output of the $\alpha$-entmax transformation applied row-wise to the score matrix $S = QK^\top$. Expressed element-wise, we obtain:

$$dV_j = \sum_{i=1}^{n} P_{ij} dO_i, \tag{17}$$

which is analogous to the softmax case. Since $P_{ij}$ is sparse due to the nature of $\alpha$-entmax, we can skip $Q_i$ blocks that leads to blocks of $P$ full of zeros using the pointer increment tables, as shown in Equation 14.

### 2. Gradient of $P$ and $S$

The next step involves computing $dP$ and $dS$. From $O = PV$, we have:

$$dP_{ij} = dO_i^\top V_j. \tag{18}$$

Next, let us recall the Jacobian of the $\alpha$-entmax mapping (Peters et al., 2019). Defining $p = \alpha\text{-entmax}(s)$, the Jacobian is:

$$\frac{\partial \alpha\text{-entmax}(s)}{\partial s} = \text{Diag}(u) - \frac{uu^\top}{\|u\|_1}, \tag{19}$$

where $\boldsymbol{u}$ is defined element-wise as:

$$u_k = \begin{cases} (p_k)^{2-\alpha}, & \text{if } p_k > 0 \\ 0, & \text{otherwise.} \end{cases} \tag{20}$$

Let $\boldsymbol{U}$ denote a stack of $[\boldsymbol{u}_1, ..., \boldsymbol{u}_n]$ for each row of $\boldsymbol{P}$. From the relationship $\boldsymbol{P} = \alpha\text{-entmax}(\boldsymbol{S})$, and the Jacobian of the $\alpha$-entmax function, we can propagate the gradients back to $\boldsymbol{S}$ as follows:

$$\boldsymbol{dS}_i = \left[ \text{Diag}(\boldsymbol{U}_i) - \frac{\boldsymbol{U}_i \boldsymbol{U}_i^\top}{\|\boldsymbol{U}_i\|_1} \right] \boldsymbol{dP}_i \tag{21}$$

$$= \boldsymbol{U}_i \odot \boldsymbol{dP}_i - \left( \frac{\boldsymbol{U}_i^\top \boldsymbol{dP}_i}{\|\boldsymbol{U}_i\|_1} \right) \boldsymbol{U}_i. \tag{22}$$

We can further simplify by defining a new quantity $\boldsymbol{\delta} \in \mathbb{R}^n$:

$$\delta_i = \frac{\boldsymbol{U}_i^\top \boldsymbol{dP}_i}{\|\boldsymbol{U}_i\|_1} \tag{23}$$

$$= \frac{1}{\|\boldsymbol{U}_i\|_1} \sum_{j=1}^{n} U_{ij} \left( \boldsymbol{dO}_i^\top \boldsymbol{v}_j \right) \tag{24}$$

$$= \boldsymbol{dO}_i^\top \underbrace{\frac{\left( \sum_{j=1}^{n} U_{ij} \boldsymbol{V}_j \right)}{\|\boldsymbol{U}_i\|_1}}_{\boldsymbol{O}_i^{(2)}} \tag{25}$$

In standard softmax attention, instead of the right-side term in the above product, we would simply obtain $\boldsymbol{O}_i$. Since this new quantity is required for the backward pass, and to avoid passing once more through $\boldsymbol{Q}$, $\boldsymbol{K}$ and $\boldsymbol{V}$, we compute and store this quantity during the forward pass solely during training. Unlike in softmax attention, however, the backward pass for $\alpha$-entmax does not require saving the output matrix $\boldsymbol{O}$; instead, we only require this new quantity, which we label $\boldsymbol{O}^{(2)}$. Then, we can simplify the computation of $\boldsymbol{dS}$ to:

$$\boldsymbol{dS}_i = \boldsymbol{U}_i \odot (\boldsymbol{dP}_i - \delta_i) \tag{26}$$

Again, we can use the sparsity stored in $\boldsymbol{M}$ (see Equation 9) from the forward pass to efficiently skip the computation of null blocks of $\boldsymbol{P}$.

### 3. Gradients of $Q$ and $K$

Using the definition of $S_{ij} = \boldsymbol{Q}_i^\top \boldsymbol{K}_j$, the gradients for $Q$ and $K$ are:

$$\boldsymbol{dQ}_i = \sum_{j=1}^{n} dS_{ij} \boldsymbol{K}_j, \tag{27}$$

$$\boldsymbol{dK}_j = \sum_{i=1}^{n} dS_{ij} \boldsymbol{Q}_i. \tag{28}$$

Substituting $dS_{ij}$, we get:

$$\boldsymbol{dQ}_i = \sum_{j=1}^{n} U_{ij} \left( dP_{ij} - \delta_i \right) \boldsymbol{K}_j \tag{29}$$

$$\boldsymbol{dK}_j = \sum_{i=1}^{n} U_{ij} \left( dP_{ij} - \delta_i \right) \boldsymbol{Q}_i \tag{30}$$

Effectively, we can only iterate through the blocks that will result in $P_{ij} \neq 0$. As in FlashAttention, the backward pass can also be computed with $\mathcal{O}(n)$ extra memory:

1. Compute $\boldsymbol{dV}_j$ for all $j$ according to Equation 17, which takes $\mathcal{O}(d)$ extra memory.

2. Compute $\delta_i$ for all $i$ according to Equation 23, which takes $\mathcal{O}(n)$ extra memory.

3. Compute $\boldsymbol{O}_i^{(2)}$ for all $i$, as defined in Equation 25, which takes $\mathcal{O}(d)$ extra memory.

4. Compute $\boldsymbol{dQ}_i$ for all $i$ according to Equation 29, which takes $\mathcal{O}(d)$ extra memory.

5. Compute $\boldsymbol{dK}_j$ for all $j$ according to Equation 30, which takes $\mathcal{O}(d)$ extra memory.

We note that the only extra memory requirement compared to FlashAttention is in having to additionally compute and storing $\boldsymbol{O}^{(2)} \in \mathbb{R}^{n \times d}$. When using block masking, we also need $\mathcal{O}(T_r \times T_c)$ extra memory to store the binary mask $\boldsymbol{M}$. However, we recall that this memory can be shared across attention layers, as it is merely a temporary matrix used to compute the pointer-increment tables.

## A.3. ADASPLASH: Forward Pass (without block masking)

The full ADASPLASH's forward pass is presented in Algorithm 2. For completeness, we also provide in Algorithm 3 the steps for approximating $\boldsymbol{\tau}$ without the need to materialize $\boldsymbol{S}$ in a block-wise manner.

---

**Algorithm 3** Halley-bisection for computing $\boldsymbol{\tau}$ – Block Version

---

**Require:** Matrices $\boldsymbol{Q}, \boldsymbol{K} \in \boldsymbol{R}^{n \times d}$ in HBM, block sizes $B_c, B_r$ and number of iterations $M$.
1: Divide $\boldsymbol{Q}$ into $T_r = \lceil n/B_r \rceil$ blocks $\boldsymbol{Q}_1, \ldots, \boldsymbol{Q}_{T_r}$ of size $B_r \times d$
2: Divide $\boldsymbol{K}$ into $T_c = \lceil n/B_c \rceil$ blocks $\boldsymbol{K}_1, \ldots, \boldsymbol{K}_{T_c}$ of size $B_c \times d$
3: Divide $\boldsymbol{\tau}$ into $T_r$ blocks $\boldsymbol{\tau}_1, \ldots, \boldsymbol{\tau}_{T_r}$ of size $B_r$
4: **for** $i = 1$ to $T_r$ **do**
5:     Load $\boldsymbol{Q}_i$ from HBM to on-chip SRAM
6:     On chip, initialize $\boldsymbol{\tau}_i, \boldsymbol{\tau}_{\text{lo}_i}, \boldsymbol{\tau}_{\text{hi}_i}$ according to Algorithm 1.          $\triangleright$ Note: this requires one pass over $\boldsymbol{K}_j$ for all $j$.
7:     **repeat**
8:         On chip, initialize $f, f', f'' = \boldsymbol{0} \in \mathbb{R}^{B_r}$
9:         **for** $j = 1$ to $T_c$ **do**
10:             Load $\boldsymbol{K}_j, \boldsymbol{V}_j$ from HBM to on-chip SRAM
11:             Compute $\boldsymbol{S}_i^{(j)} = \boldsymbol{Q}_i \boldsymbol{K}_j^\top \in \mathbb{R}^{B_r \times B_c}$
12:             Accumulate $f, f', f''$ according to Equations 3, 6 and 7, respectively.
13:         **end for**
14:         Update $\boldsymbol{\tau}_i, \boldsymbol{\tau}_{\text{lo}_i}, \boldsymbol{\tau}_{\text{hi}_i}$ according to Algorithm 1.
15:     **until** $M$ iterations are completed
16:     Write $\boldsymbol{\tau}_i$ to HBM
17: **end for**
18: **Return:** $\boldsymbol{\tau}$

---

## A.4. ADASPLASH: Backward Pass (without block masking)

As mentioned in §3.2.2, in contrast to FlashAttention, we propose to separate the kernels that compute the gradients $\boldsymbol{dQ}, \boldsymbol{dK}, \boldsymbol{dV}$. However, as in FlashAttention, we need to compute $\boldsymbol{\delta}$ before being able to compute the gradients, which we do in a separate kernel following Equation 25. We present the full steps for computing $\boldsymbol{dK}$ and $\boldsymbol{dV}$ in Algorithm 4, and for computing $\boldsymbol{dQ}$ in Algorithm 5.

## A.5. ADASPLASH: Block Masked Version

In this version, as outlined in Section 3, a boolean block mask $\boldsymbol{M} \in \mathbb{R}^{T_r \times T_c}$ is created dynamically in the forward pass, allowing the exploitation of the sparsity in the matrix $\boldsymbol{P}$ at the cost of linear memory complexity. The mask is populated

**Algorithm 4** ADASPLASH Backward Pass for $dK$ and $dV$

---

**Require:** Matrices $Q, K, V, O, dO \in \mathbb{R}^{n \times d}$ in HBM, vector $\boldsymbol{\tau} \in \mathbb{R}^n$ in HBM, block sizes $B_c, B_r$, parameter $\alpha$

1: Divide $Q$ into $T_r = \lceil n/B_r \rceil$ blocks $Q_1, \ldots, Q_{T_r}$ of size $B_r \times d$ each, and divide $K, V$ into $T_c = \lceil n/B_c \rceil$ blocks $K_1, \ldots, K_{T_c}, V_1, \ldots, V_{T_c}$ of size $B_c \times d$ each.

2: Divide $dO$ into $T_r$ blocks $dO_1, \ldots, dO_{T_r}$ of size $B_r \times d$ each.

3: Divide $\boldsymbol{\tau}$ into $T_r$ blocks $\boldsymbol{\tau}_1, \ldots, \boldsymbol{\tau}_{T_r}$ of size $B_r$ each.

4: Initialize and divide $dK, dV \in \mathbb{R}^{n \times d}$ into $T_c$ blocks $dK_1, \ldots, dK_{T_c}$ and $dV_1, \ldots, dV_{T_c}$ of size $B_c \times d$ each.

5: Divide $\boldsymbol{\delta}$ into $T_r$ blocks $\boldsymbol{\delta}_1, \ldots, \boldsymbol{\delta}_{T_r}$ of size $B_r$ each.

6: **for** $1 \le j \le T_c$ **do**

7:      Load $K_j, V_j$ from HBM to on-chip SRAM.

8:      Initialize $dK_j = \mathbf{0}_{B_c \times d}$ on SRAM.

9:      Initialize $dV_j = \mathbf{0}_{B_c \times d}$ on SRAM.

10:      **for** $1 \le i \le T_r$ **do**

11:          Load $Q_i, dO_i, \boldsymbol{\tau}_i, \boldsymbol{\delta}_i$ from HBM to on-chip SRAM.

12:          On chip, compute $S_i^{(j)} = Q_i K_j^\top \in \mathbb{R}^{B_r \times B_c}$.

13:          On chip, compute $P_i^{(j)} = \max(0, (\alpha - 1) S_i^{(j)} - \boldsymbol{\tau}_i)^{1/\alpha - 1} \in \mathbb{R}^{B_r \times B_c}$.

14:          On chip, compute $dV_j \leftarrow dV_j + (P_i^{(j)})^\top dO_i \in \mathbb{R}^{B_c \times d}$.

15:          On chip, compute $dP_i = dO_i V_j^\top \in \mathbb{R}^{B_r \times B_c}$.

16:          On chip, compute $U_i^{(j)} = P_i^{(j)^{2-\alpha}} \in \mathbb{R}^{B_r \times B_c}$.

17:          On chip, compute $dS_i^{(j)} = U_i^{(j)} \odot (dP_i^{(j)} - \boldsymbol{\delta}_i) \in \mathbb{R}^{B_r \times B_c}$.

18:          On chip, compute $dK_j \leftarrow dK_j + (dS_i^{(j)})^\top Q_i \in \mathbb{R}^{B_c \times d}$.

19:      **end for**

20:      Write $dK_j, dV_j$ to HBM.

21: **end for**

22: **Return:** Gradients $dK, dV$.

---

during the final iteration of the Halley-bisection algorithm (Algorithm 3) by evaluating the condition $\text{any}(S_i^{(j)} > \boldsymbol{\tau}_i)$ and storing the result as a boolean value. Thus, the mask $M$ indicates whether a specific $Q, K$ block pair contributes to the output. This process enables the creation of a lookup table that associates each query block with the set of key blocks that contribute non-zero values, thereby allowing to skip unnecessary computations for future computations. Similarly, a reverse lookup table can be created for each key block. Both tables can be used in the backward pass (Line 10 in Algorithm 4 and Line 9 in Algorithm 5) to avoid looping over unnecessary query/key blocks.

In practice, to create the lookup tables, we use the `torch.argwhere` function to extract the $(i, j)$ indices of entries where $M_{ij} = 1$. Combined with row-wise summation of non-zero entries, this approach efficiently skips computations for irrelevant blocks within the remaining kernels. Consequently, during the forward pass, only the $K, V$ pairs identified in the lookup table are loaded, avoiding redundant memory and computational overhead. As mentioned, for the backward pass, given that we separated the computation of $dQ$ and $dK, dV$, we can further use both tables ($\mathcal{Q}$ and $\mathcal{K}$) to speedup the gradient computation.

## B. Experimental Setup

### B.1. Continuous Pre-training

We conducted continuous pretraining of RoBERTa-base[3] and ModernBERT-base[4] models with our custom sparse attention Triton kernel, ADASPLASH. The pretraining process was carried on 2B tokens of the FineWeb-Edu dataset,[5] due to its high-quality, diverse and large-scale content. We used the HuggingFace Transformers library for model training and implementation and the Datasets library for data handling. Concretely, we used a batch size of 32 and a learning rate of $5 \times 10^{-5}$, optimized with the AdamW optimizer. Training was conducted for 100,000 steps using mixed-precision (`fp16`).

---

[3] https://huggingface.co/FacebookAI/roberta-base
[4] https://huggingface.co/answerdotai/ModernBERT-base
[5] https://huggingface.co/datasets/HuggingFaceFW/fineweb-edu

---

**Algorithm 5** ADASPLASH Backward Pass for $dQ$

---

**Require:** Matrices $Q, K, V, O, dO \in \mathbb{R}^{n \times d}$ in HBM, vector $\tau \in \mathbb{R}^n$ in HBM, block sizes $B_c, B_r$, parameter $\alpha$.

1: Divide $Q$ into $T_r = \lceil n/B_r \rceil$ blocks $Q_1, \ldots, Q_{T_r}$ of size $B_r \times d$ each, and divide $K, V$ into $T_c = \lceil n/B_c \rceil$ blocks $K_1, \ldots, K_{T_c}, V_1, \ldots, V_{T_c}$ of size $B_c \times d$ each.
2: Divide $dO$ into $T_r$ blocks $dO_1, \ldots, dO_{T_r}$ of size $B_r \times d$ each.
3: Divide $\tau$ into $T_r$ blocks $\tau_1, \ldots, \tau_{T_r}$ of size $B_r$ each.
4: Initialize $dQ$ in HBM and divide it into $T_r$ blocks $dQ_1, \ldots, dQ_{T_r}$ of size $B_r \times d$ each.
5: Divide $\delta$ into $T_r$ blocks $\delta_1, \ldots, \delta_{T_r}$ of size $B_r$ each.
6: **for** $i = 1$ to $T_r$ **do**
7:     Load $Q_i, dO_i, \delta_i, \tau_i$, from HBM to on-chip SRAM
8:     Initialize $dQ_i = \mathbf{0}_{B_c \times d}$ on SRAM.
9:     **for** $j = 1$ to $T_c$ **do**
10:         On chip, compute $S_i^{(j)} = Q_i K_j^\top \in \mathbb{R}^{B_r \times B_c}$.
11:         On chip, compute $P_i^{(j)} = \max(0, (\alpha - 1)S_i^{(j)} - \tau_i)^{1/\alpha - 1} \in \mathbb{R}^{B_r \times B_c}$.
12:         On chip, compute $dP_i = dO_i V_j^\top \in \mathbb{R}^{B_r \times B_c}$.
13:         On chip, compute $U_i^{(j)} = P_i^{(j)2-\alpha} \in \mathbb{R}^{B_r \times B_c}$.
14:         On chip, compute $dS_i^{(j)} = U_i^{(j)} \odot (dP_i^{(j)} - \delta_i) \in \mathbb{R}^{B_r \times B_c}$.
15:         On chip, compute $dQ_i \leftarrow dQ_i + dS_i^{(j)} K_j \in \mathbb{R}^{B_r \times d}$.
16:     **end for**
17:     Write $dQ_i$ to HBM
18: **end for**
19: **Return:** Gradient $dQ$

---

*Table 5.* Runtime (s) of ModernBERT-base ($\alpha = 1.5$) for varying context lengths.

| | Sequence Length | | | | |
|---|---|---|---|---|---|
| Algorithm | 512 | 1024 | 2048 | 4096 | 8192 |
| Sorting (Torch) | 0.09 | 0.11 | 0.26 | 0.76 | OOM |
| Bisection (Torch) | 0.11 | 0.15 | 0.42 | 1.35 | 4.99 |
| Halley-bisection (Triton) | 0.10 | 0.11 | 0.26 | 0.46 | 1.61 |
| ADASPLASH (Triton) | 0.10 | 0.12 | 0.21 | 0.48 | 1.53 |

The sparsity parameter ($\alpha$) was initialized at 1.01 and annealed linearly to a final value of 1.5 or 2.0 over 50,000 steps. We kept ModernBERT's window attention layers untouched, only replacing the full softmax layers by $\alpha$-entmax. Finally, we also performed continuous pretraining of RoBERTa and ModernBERT with standard softmax attention with a fixed $\alpha = 1.0$.

As shown in Figure 5, the attention mechanisms of our sparse ModernBERT model ($\alpha = 1.5$) obtain high sparsity levels in practice, with an overall sparsity of 95% for $\alpha = 1.5$ and 99% for $\alpha = 2.0$. For this reason, we used the version of ADASPLASH that leverages the pointer increment tables for training ModernBERT, which has a maximum sequence length of 8,192. For RoBERTa, which has a sequence length of 512, we opted to use the Halley-bisection algorithm implemented in Triton. In Table 5 we report efficiency results in terms of runtime and memory usage for different attention algorithms with ModernBERT-base. Overall, we observe that the sorting approach is slower than bisection, which is slower than our Halley-bisection and ADASPLASH, in that order.

## B.2. GLUE and BIER tasks

For GLUE tasks, we used the checkpoints of continuous pre-trained models for both RoBERTa-base and ModernBERT-base. Then, we fine-tuned them on each GLUE task with the default hyperparameters from the Transformer library.[6] Importantly, we capped the maximum sequence length at 128 tokens to reduce computational cost while preserving task-relevant context and used fp16 for training.

---

[6]https://github.com/huggingface/transformers/tree/main/examples/pytorch/text-classification

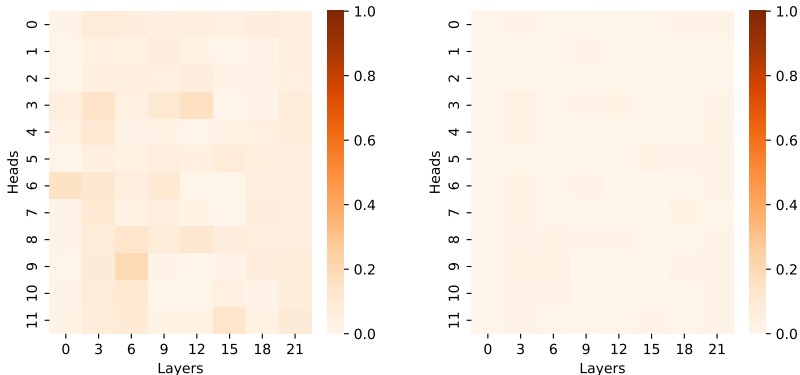

*Figure 5.* Ratio of non-zeros for non-local layers of ModernBERT-base with $\alpha = 1.5$ (left) and $\alpha = 2.0$ (right).

*Table 6.* Results on different tasks from the GLUE benchmark (Wang et al., 2018).

| Model | Params | Seq. | Single Sentence | | Paraphrase and Similarity | | | Natural Language Inference | | | |
| | | | CoLA | SST-2 | MRPC | STS-B | QQP | MNLI | QNLI | RTE | Avg. |
|---|---|---|---|---|---|---|---|---|---|---|---|
| BERT | 110M | 512 | 58.6 | 91.9 | 86.9 | 89.0 | 89.3 | 84.0 | 91.0 | 69.3 | 82.5 |
| RoBERTa | 125M | 512 | 59.8 | 93.7 | 89.5 | 89.6 | 89.8 | 87.7 | 92.3 | 69.3 | 83.9 |
| RoBERTa ($\alpha = 1.5$) | 125M | 512 | 58.5 | 93.2 | 91.5 | 90.2 | 89.7 | 87.3 | 92.5 | 68.6 | 83.9 |
| RoBERTa ($\alpha = 2.0$) | 125M | 512 | 56.8 | 93.0 | 90.9 | 88.8 | 89.0 | 86.7 | 91.9 | 67.2 | 83.0 |
| ModernBERT | 149M | 8192 | 63.2 | 95.0 | 88.2 | 90.3 | 90.4 | 87.9 | 93.0 | 61.7 | 83.7 |
| ModernBERT ($\alpha = 1.5$) | 149M | 8192 | 62.2 | 96.1 | 87.7 | 89.4 | 90.2 | 87.9 | 92.6 | 61.7 | 83.5 |
| ModernBERT ($\alpha = 2.0$) | 149M | 8192 | 62.2 | 94.8 | 89.0 | 89.9 | 90.5 | 87.8 | 93.1 | 62.5 | 83.7 |

To evaluate the generalization of ADASPLASH in retrieval tasks, we fine-tuned ModernBERT-base and RoBERTa-base models on the MS MARCO dataset (Bajaj et al., 2016) and evaluated them on the BEIR benchmark (Thakur et al., 2021). This benchmark suite assesses performance across diverse information retrieval tasks, including SciFact, NFCorpus, FiQA-2018, and TREC-COVID. The fine-tuning and evaluation process closely follows the approach proposed in the ModernBERT paper (Warner et al., 2024). Fine-tuning was performed using the SentenceTransformers library.[7] The models were evaluated on BEIR tasks using the MTEB benchmark toolkit.[8] The evaluation metric for each task was nDCG@10 (Normalized Discounted Cumulative Gain), following standard information retrieval practices.

### B.3. Long Document Classification

The European Court of Human Rights (ECtHR) dataset comprises legal cases from the European Court of Human Rights, each associated with specific articles of the Convention on Human Rights allegedly violated. For this task, we fine-tuned the RoBERTa base model (Liu et al., 2019) with a classification head. Since this is a multi-label classification task, we used a binary cross-entropy loss. To accommodate longer contexts, we followed the approach proposed by (Beltagy et al., 2020), repeating the 512 position embeddings until the target context size was reached. We used the AdamW optimizer for training. For hyperparameters, we follow the recipe of Dai et al. (2022). For the attention mechanism, `bfloat16` precision was used.

### B.4. Language Modeling

We trained both the standard GPT-2 model and sparse GPT-2 ($\alpha = 1.5$) using the configuration provided in the `llm.c` repository,[9] following their training recipe. Specifically, we trained a GPT-2 (124M parameters) from scratch on 10B tokens of the FineWeb dataset, with a maximum sequence length of 1024 tokens. Training was conducted using `bfloat16` precision. We use an effective batch size of 512, and use gradient accumulation to fit into available GPU memory. We

---

[7] https://sbert.net/
[8] https://github.com/embeddings-benchmark/mteb
[9] https://github.com/karpathy/llm.c

use the AdamW optimizer, with learning rate $6 \times 10^{-4}$ and weight decay of 0.1. The learning rate followed a warm-up phase, linearly ramping from zero to a maximum of $6 \times 10^{-4}$ over the first 700 iterations, equivalent to 350 million tokens. Subsequently, the learning rate decayed to zero across the remaining training steps. We show the validation loss curves for both softmax and $\alpha$-entmax ($\alpha = 1.5$) in Figure 6.

Given that, for this task, the context size was not high enough, for sparse attention we opted to use the algorithm that does not take advantage of the pointer increment tables. For the benchmarking of the time spent per step, we averaged across 50 steps after the model had trained for at least 100 steps.

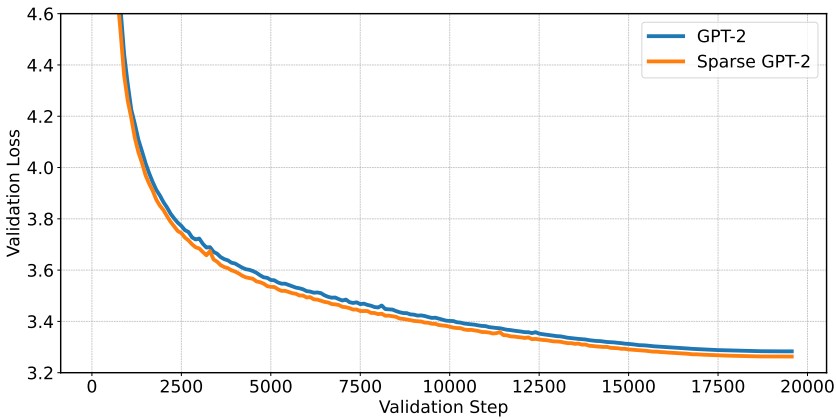

*Figure 6.* FineWeb withheld validation loss comparison between GPT-2 and Sparse GPT-2 during training.

## C. Computational Details

Experiments on masked language modeling, text classification, GLUE tasks and BIER tasks were carried on Nvidia RTX A6000 GPUs with 48GB VRAM. Experiments with GPT-2 and the efficiency benchmark in Figures 1 and 3 were carried on a single Nvidia H100 GPU (80GB). The runtime experiments with ModernBERT were carried on a single A6000 GPU.

