# OpenReview forum: "AdaSplash: Adaptive Sparse Flash Attention"
_ICML.cc/2025/Conference — ICML 2025 oral_

### Official Review · Reviewer_bvB9 · 2025-03-02

**Overall Recommendation:** 3

**Summary:**

This paper aims to improve the efficiency of alpha-entmax attention, which is a kind of adaptive sparse attention. Its contributions are two folds:

1. Alpha-entmax attention requires a threshold to determine which tokens are masked. This paper proposes an algorithm that achieves faster estimation of this threshold.
2. After the threshold is estimated, this paper follows the spirit of Flash Attention and develops a hardware-optimized implementation of alpha-entmax attention called Sparse Flash Attention. The key idea is to avoid materializing the quadratic similarity matrices.

Experiments demonstrate significant speedups compared with previous implementations of alpha-entmax attention.

**Claims And Evidence:**

Claim 1: This method achieves substantial computational improvements over existing alpha-entmax implementations.
This is supported by clear and convincing evidence.

Claim 2: This method can often match or even surpass the efficiency of highly optimized softmax-based attention algorithms like FlashAttention-2.
This is supported by the experiments. When the block-sparsity higher than about 95%, the proposed method is faster than FlashAttention-2.

**Essential References Not Discussed:**

This paper improves alpha-entmax attention, which is a special case of sparse attention.
The authors should discuss the pros and cons of alpha-entmax attention compared with other types of sparse attention, such as topk attention [1] and block sparse attention.

[1] Memory-efficient Transformers via Top-k Attention, ACL2021

**Experimental Designs Or Analyses:**

I check the speed comparison part (figure 1,2 and 3).
The authors only evaluate the running speed on random data (figure 3). They should also evaluate the running speed on real dataset.
Moreover, the author should report the running time of Halley-bisection alone.

I check the accuracy comparison part (table 1, 2 and 3).
The results support the authors claims. But I am not familiar with the benchmark in table 1.

**Methods And Evaluation Criteria:**

The proposed methods and evaluation criteria make sense.

**Other Comments Or Suggestions:**

None

**Other Strengths And Weaknesses:**

Weaknesses:
This paper focus on the efficiency of sparse attention, while Flash-attention is a dense attention. When compared with Flash-attention, we would expect that the proposed method is much faster than Flash-attention due to the sparsity. However, their speeds are very close. This limits practical applications of the proposed method—why would one use the proposed method instead of Flash-attention?

My biggest concern is that the estimation of threshold requires multiple access of QK memory. Thus it is difficult to be faster than FlashAttention. Online estimation might be better in term of speed.

**Questions For Authors:**

None

**Relation To Broader Scientific Literature:**

Flash attention /
Sparse attention /
alpha-entmax /
Halley’s method for faster bisection convergence

**Theoretical Claims:**

I carefully check the correctness of algorithm 1 and algorithm 2.
I don't check the correctness of the backward pass algorithm.

---

> ### Author Rebuttal · Authors · 2025-03-31
>
> Thank you for your insightful comments. We are glad that you found our paper provides clear and convincing evidence about the computational improvements of our implementations, and that our proposed methods and evaluation criteria make sense. We address your main concerns below.
>
>
> **“They should also evaluate the running speed on real dataset. Moreover, the author should report the running time of Halley-bisection alone.”**
>
> Regarding the runtime of the Halley-bisection algorithm, please note that we **do** report these results in Section 3.1 under the _Efficiency Benchmark_ paragraph (Line 182).
>
> For the AdaSplash implementation, we followed FlashAttention-2 and chose to use synthetic data to evaluate our kernels, which allows us to control sparsity levels and systematically explore performance across varying context lengths (see Figures 1 and 3). Note, however, that we **do** report runtime on real datasets in our end-to-end training benchmarks (Table 3, 4, and 6) comparing the overall speed of models using AdaSplash with those using FlashAttention-2.
>
>
>
> **“This paper improves alpha-entmax attention, which is a special case of sparse attention. The authors should discuss the pros and cons of alpha-entmax attention compared with other types of sparse attention, such as topk attention [1] and block sparse attention.”**
>
> This is a good suggestion, which we will follow. As also noted in our response to Reviewer CoRN, methods like top‑k and block-sparse attention rely on fixed, input-agnostic sparsity patterns, which may miss important tokens. Top‑k requires setting k in advance and block-sparse methods depend on a pre-defined mask and cannot adapt to specific patterns for each query vector. In contrast, α-entmax offers adaptive sparsity, selecting relevant tokens based on the input without requiring hard thresholds or fixed mask patterns. Its sparsity emerges naturally from the transformation, which is **exact**, **differentiable**,  and it generalizes softmax (when $\alpha \to 1$) without any approximations.
>
> We will further clarify these distinctions in the final version.
>
>
>
> **“When compared with Flash-attention, we would expect that the proposed method is much faster than Flash-attention due to the sparsity. [..] why would one use the proposed method instead of Flash-attention?”**
>
> We appreciate the reviewer’s concern regarding efficiency. As stated above, AdaSplash introduces adaptive, unstructured sparsity, which contrasts with fixed sparsity patterns that intentionally trade accuracy for speed. This makes implementation and optimization more challenging, but also more expressive and flexible.
>
> Our primary contribution is accelerating α-entmax attention, rather than introducing a generic sparse attention mechanism that merely approximates softmax attention. AdaSplash is the first efficient implementation of α-entmax, and already achieves speeds comparable to FlashAttention, despite being at an earlier stage of hardware/software optimization.
> In practice, we have seen in Figures 4-5 that sparsity emerges in real attention patterns. Beyond speed, our method may also improve task performance and generalization, particularly for long-context scenarios where it may help to mitigate attention dispersion, a problem known to affect softmax-based dense attention [1, 2]. This may also explain why AdaSplash performs strongly across multiple benchmarks (Tables 1–3), especially in retrieval tasks.
>
>
>
>
>
> **“My biggest concern is that the estimation of threshold requires multiple access of QK memory. Thus it is difficult to be faster than FlashAttention. Online estimation might be better in term of speed.”**
>
> It is true that, unlike FlashAttention which reads $Q, K, V$ tensors once in the forward pass, AdaSplash requires at least an additional read on $k$ and an additional $QK^\top$ block computation. However, thanks to potential block-wise sparsity in α-entmax, we can skip entire blocks of $V$ (sidestepping a slow memory read) and avoid computing their contribution during the $PV$ step in the forward pass. Similarly, during the backward pass, masked blocks are skipped entirely, which makes, in practice, AdaSplash’s backward pass faster than FlashAttention’s. However, we do emphasize that this ultimately depends on the sparsity (as per Figure 1).
>
> As for online estimation, we agree it's an interesting direction for future work. Unlike softmax, estimating the threshold $\tau$ for α-entmax requires full access to the input, making online computation more challenging.  Thanks for the suggestion!
>
>
>
> References:
>
>
> [1] Veličković, Petar, et al. "softmax is not enough (for sharp out-of-distribution)." arXiv preprint arXiv:2410.01104 (2024).
>
>
> [2] Ye, Tianzhu, et al. "Differential transformer." arXiv preprint arXiv:2410.05258 (2024).

---

> > ### Comment · Reviewer_bvB9 · 2025-04-09
> >
> > The authors' rebuttal tackles part of my concerns. My biggest concern is that the proposed method should be much faster than FlashAttention but it is not. But after reading the rebuttal and re-considering the value of this work, I think it is not necessary to be much faster than FlashAttention. As the authors claimed, this work has its own values.
> >
> > So I improve my score from 2 to 3. And I have an additional question: could you please list the number of Q/K/V reads as a table?

---

### Official Review · Reviewer_agna · 2025-03-13

**Overall Recommendation:** 5

**Summary:**

The paper introduces ADASPLASH, an efficient implementation of α-entmax attention that leverages sparsity to improve computational performance while maintaining model quality. The key contributions are:

1. A hybrid Halley-bisection algorithm that reduces the number of iterations needed to compute α-entmax transformation
2. Custom Triton kernels that efficiently handle adaptive sparsity in both forward and backward passes

**Claims And Evidence:**

The primary claims are well-supported by evidence

**Essential References Not Discussed:**

None.

**Experimental Designs Or Analyses:**

The experimental design is thorough and well-executed:

1. Comprehensive comparison against multiple baselines (FlashAttention-2 CUDA, FlashAttention-2 Triton, Torch bisection)

2. Testing across multiple sequence lengths (512 to 64k tokens)

3. Evaluation on diverse tasks and architectures

4. Appropriate ablation studies (masked vs. non-masked variants)

**Methods And Evaluation Criteria:**

The methods are appropriate and well-justified.

**Other Comments Or Suggestions:**

None.

**Other Strengths And Weaknesses:**

Strengths:

The approach is architecture-agnostic, demonstrating benefits on both encoder and decoder models.

Weaknesses:

The improvements are most significant at high sparsity levels (>90%), which may not be reached in all applications.
The efficiency gains for decoder-only models (GPT-2) are more modest than for encoder models, suggesting varying utility across architectures.
The approach still requires materializing the mask matrix M with O(Tr × Tc) complexity, which could become a bottleneck for extremely long sequences.

**Questions For Authors:**

Can the proposed method reuse the memory of KV with zero attention scores? If not, can the method actually save memory?

**Relation To Broader Scientific Literature:**

1. Adaptive sparse attention mechanisms
2. Efficient attention implementations

**Theoretical Claims:**

The mathematical derivations for both the Halley-bisection algorithm and the attention mechanism appear sound. The derivation of α-entmax's Jacobian and its sparse properties (Section 3.2.2) is correctly presented and consistent with prior work. The block-wise computation approach in Algorithm 1 and the memory complexity analysis in Sections 3.2.1 and 3.2.2 are mathematically sound.

---

> ### Author Rebuttal · Authors · 2025-03-31
>
> Thank you for your positive comments. We address your points below.
>
> **“Can the proposed method reuse the memory of KV with zero attention scores? If not, can the method actually save memory?”**
>
> Similarly to FlashAttention, AdaSplash is primarily optimized for efficient training. For inference settings, for example, other methods such as Flash Decoding (an extension of FlashAttention) are needed in order to explicitly optimize inference by splitting the KV-cache memory and at the end aggregating the results. Because our work exclusively focuses on the training scenario, we have not addressed KV-cache optimizations. Nevertheless, investigating the potential benefits of leveraging the sparsity from α-entmax to optimize KV-cache utilization is an orthogonal and promising direction for future work.
>
>
> In the training regime, compared to naive/eager attention implementation, AdaSplash significantly reduces memory usage which can be seen by Figure 3 as well as Tables 3 and 6 where the eager method goes out-of-memory for bigger contexts. We do this by avoiding the explicit materialization of large intermediate matrices such as the attention score matrix $S$ and the attention probability matrix $P$. We show that, just like in FlashAttention, one can avoid storing these matrices which in turn both reduces the memory usage and speeds up the computation.
>
>
> **“The improvements are most significant at high sparsity levels (>90%), which may not be reached in all applications. The efficiency gains for decoder-only models (GPT-2) are more modest than for encoder models, suggesting varying utility across architectures.”**
>
> Interestingly, in the specific case of GPT-2, our analysis (Figure 4) reveals a clear pattern where the first attention layer prefers denser attention distributions, while subsequent layers become significantly sparse. This observation naturally suggests the possibility of using an hybrid approach, combining dense attention at earlier layers with AdaSplash attention at later layers in order to potentially obtain the best balance between computation efficiency and model performance. We plan to incorporate this analysis with GPT-2 in the final version.
>
> **“The approach still requires materializing the mask matrix M with O(Tr × Tc) complexity, which could become a bottleneck for extremely long sequences.”**
>
> You are right to note the memory overhead involved in materializing the block mask. Thus, we provide two implementations of AdaSplash: one that explicitly stores a binary block-sparsity mask (adding some memory overhead), and another without explicit mask storage (matching the memory requirements of FlashAttention-2) in subsection 3.2 (Line 299). For our experiments, the block mask memory overhead was not a bottleneck (Table 6). Moreover, we stress tested our implementation up to 64K context length and did not observe any memory issues, mainly because the matrix $M$ stores single bytes – extremely memory efficient in practice.

---

### Official Review · Reviewer_CoRN · 2025-03-16

**Overall Recommendation:** 3

**Summary:**

The paper introduces ADASPLASH, a novel method that combines the computational efficiency of GPU-optimized algorithms with the sparsity benefits of the α-entmax attention family. The goal is to enable efficient and scalable training of transformers for long-context tasks while maintaining or improving task performance.

## update after rebuttal

The author's rebuttal addressed most of my concerns; I will keep my score.

**Claims And Evidence:**

Yes, the authors first propose a hybrid Halley-bisection algorithm for faster computation of the α-entmax transformation, reducing the number of iterations by 7-fold compared to standard methods. Then, to utilize this sparse pattern, the authors propose adaptive sparsity-aware Triton kernels to exploit sparsity in both the forward and backward passes of attention computations.

**Essential References Not Discussed:**

None

**Experimental Designs Or Analyses:**

The experiments report evaluation results under different alpha values but do not provide corresponding computational efficiency metrics, such as sparsity levels and computation time. For instance, in Table 1, the 8192-length setting, which is the longest in the paper, could effectively showcase the efficiency advantages of sparsity but lacks such analysis.

**Methods And Evaluation Criteria:**

The experiments are primarily conducted on BERT and GPT-2 level models, without testing on larger-scale models such as 7B-scale LLMs. Additionally, evaluations on long-text benchmarks, such as Needle in a Haystack and RULER, are missing. Therefore, it is unclear whether this sparse attention mechanism can effectively handle mainstream long-text tasks.

**Other Comments Or Suggestions:**

None

**Other Strengths And Weaknesses:**

None

**Questions For Authors:**

- What is the comparison of computational efficiency between softmax and α-entmax, including both theoretical time complexity and actual computation time? Does Figure 1 include the full computation time, or only the time after applying the sparse mask?
- What is the relationship between the α parameter and sparsity? Does a larger α always result in greater sparsity? Is there a theoretical basis or empirical evidence supporting this relationship? Additionally, does the block size setting in the Triton kernel affect the sparsity of the sparse mask? If so, what is the nature of this effect?
- Compared to traditional fixed sparsity patterns, such as selecting tokens with a stride, what are the main advantages of adaptive sparsity? Are there any empirical comparisons to illustrate these advantages?

**Relation To Broader Scientific Literature:**

The paper's key contributions build on prior work in sparse attention mechanisms and efficient transformer architectures:

- Relation to α-entmax: It extends the α-entmax family by introducing a more efficient Halley-Bisection algorithm, significantly improving its computational feasibility for large-scale tasks.
- Relation to Efficient Attention: It complements methods like Longformer and FlashAttention by leveraging adaptive sparsity, offering dynamic attention patterns instead of fixed sparsity structures.
- Hardware-Aware Optimization: The introduction of sparsity-aware Triton kernels advances hardware-optimized transformer implementations, bridging theoretical improvements with practical GPU efficiency.

**Theoretical Claims:**

N/A

---

> ### Author Rebuttal · Authors · 2025-03-31
>
> Thank you for your insightful comments and suggestions.
>
> **“The experiments are primarily conducted on BERT and GPT-2 level models, without testing on larger-scale models such as 7B-scale LLMs. Additionally, evaluations on long-text benchmarks, such as NIAH and RULER, are missing...”**
>
> Please note that our primary goal is to introduce AdaSplash, an efficient implementation of entmax attention [1], rather than to benchmark sparse vs. dense attention on long-text tasks or large-scale models. While we don’t include 7B-scale evaluations or long-text benchmarks, we address a clear gap: entmax lacked an efficient implementation. As a bonus, we show that AdaSplash can even outperform FlashAttention in wall-clock speed (Fig. 1 & 3). Like FlashAttention, which also omitted large-scale retrieval experiments, we believe AdaSplash is a fundamental tool to enable future research in these directions.
>
>
> **“The experiments report evaluation results under different alpha values but do not provide corresponding computational efficiency metrics, such as sparsity levels and computation time....”**
>
> We **do** provide key runtime and memory metrics in Tables 3, 4, and 6 in the Appendix. In particular, Table 6 specifically shows runtime for a 8192-token context using RoBERTa. We'll integrate and highlight these results more clearly in the final version.
>
>
> **“What is the comparison of computational efficiency between softmax and α-entmax, including both theoretical time complexity and actual computation time?”**
>
> In theory, α-entmax requires slightly more work than softmax in the forward pass – about $(2 + k)N$ vs. $2N$ – but we considerably reduce k to just 2 or 3 iterations using a hybrid Halley-Bisection method, greatly improving runtime compared to previous methods (Fig. 2). When used in attention mechanisms, the main efficiency gains of α-entmax come in: (1) avoiding the multiplication by $V$ in the forward pass and (2) skipping zero blocks in the backward pass, where α-entmax has sublinear complexity (see Eq. 12-14). Note that AdaSplash exploits both points for faster computation.
>
>
> **“Does Figure 1 include the full computation time, or only the time after applying the sparse mask?”**
>
> Figure 1 reports the **full** computation time of forward + backward passes, and thus, it already includes both the computation for finding $\tau$ and producing the output.
>
>
> **“What is the relationship between the α parameter and sparsity? Does a larger α always result in greater sparsity? Is there a theoretical basis or empirical evidence supporting this relationship?”**
>
> Indeed, as discussed in line 92, there is a direct theoretical relationship between α and sparsity: for the same input vector, increasing α results in an equal or greater sparsity of the output attention distribution. This theoretical result, along with empirical evidence can be found in the original entmax paper [1, Table 3].
>
>
> **“Additionally, does the block size setting in the Triton kernel affect the sparsity of the sparse mask? If so, what is the nature of this effect?”**
>
> Indeed, the block size affects the sparsity of the block mask. Since masking is done at the block level (a block is skipped only if all its elements are zero), larger blocks reduce the chance of being fully masked. Conversely, smaller blocks increase the likelihood of full sparsity and improve computational efficiency but might induce a slightly higher memory overhead to store the bit mask (see response to R2 for more details on the memory cost).
>
> The table below illustrates how block-wise sparsity varies with different block sizes (row: $B_r$, column: $B_c$) for a fixed input of 32k context length:
> | | 16   | 32   | 64   | 128  |
> |--|--|--|--|--|
> | **16**  | 96.8 | 93.6 | 87.2 | 76.1 |
> | **32**  | 93.6 | 87.7 | 76.4 | 58.4 |
> | **64**  | 87.7 | 77.0 | 59.3 | 35.3 |
> | **128** | 78.1 | 61.0 | 37.4 | 14.2 |
>
> Finally, while block size impacts practical efficiency, it does not affect the mathematical sparsity produced by the α-entmax function, which depends solely on α and the inputs.
>
>
>
>
> **“Compared to traditional fixed sparsity patterns, such as selecting tokens with a stride, what are the main advantages of adaptive sparsity?”**
>
> Adaptive sparsity offers more flexibility than fixed patterns by dynamically selecting the most relevant tokens based on the input, avoiding the risk of overlooking important ones. Unlike fixed sparsity, which approximates softmax for efficiency, α-entmax is an exact, smooth transformation that naturally produces sparse outputs, allowing computational savings without sacrificing accuracy. Moreover, prior work (Treviso et al., 2021) has shown empirically that α-entmax outperforms fixed-pattern attention methods (e.g., BigBird, Longformer, Reformer) in tasks like machine translation and masked language modeling, especially under high sparsity conditions.
>
>
> References:
>
> [1] Peters, B., Niculae, V., & Martins, A. F. “Sparse sequence-to-sequence models.” ACL 2019.

---

### Decision · Program_Chairs · 2025-05-01

**Decision:**

Accept (oral)

**Comment:**

Overall, I think this is a solid contribution with the following strengths:
- It provides an efficient algorithm and hardward-aware implementation of an adaptive input-dependent sparse attention.
- This sparse attention mechanism is thoroughly evaluated across various applications and context lengths, highlighting that the proposed adasplash for sparse attention can match the efficiency of FlashAttention, while providing minor predictive performance improvements. The evaluation appears quite thorough in terms of evaluating the attention mechanism across various different scenarios.

The main limitation appears to be that this mechanism has an additional memory overhead for really long context lengths, and can only at best match FlashAttention, though the one of promises of sparsity is the ability to be faster than the quadratic cost of dense attention. However, the authors mention that the memory overhead can be quite negligible even at 64k context length as the mask matrix is a binary matrix. The speedup over dense attention is only expected when the sparsity levels would be significantly large, which is a problem dependent property as $\alpha$-entmax sparse attention sets the attention sparsity in a problem dependent manner instead of enforcing a certain level of sparsity.